# Activation of Peroxymonosulfate Using Spent Li-Ion Batteries for the Efficient Degradation of Chloroquine Phosphate

**Zhenzhong Hu [1,2,\*], Jia Luo [1,2], Sheng Xu [1,2], Peng Yuan [3,\*], Shengqi Guo [1,2], Xuejing Tang [1,2] and Boxiong Shen [1,2,\*]**

1 Hebei Engineering Research Center of Pollution Control in Power System, Tianjin 300401, China
2 Tianjin Key Laboratory of Clean Energy and Pollutant Control, School of Energy and Environmental Engineering, Hebei University of Technology, Tianjin 300401, China
3 Tianjin Recyclable Resources Institute, All China Federation of Supply and Marketing Cooperatives, Tianjin 300191, China
\* Correspondence: huzhenzhong@hebut.edu.cn (Z.H.); yuanpeng@rrtj.cn (P.Y.); shenbx@hebut.edu.cn (B.S.)

**Abstract:** Recycling and reusing spent lithium-ion batteries (LIBs) have gained a lot of attention in recent years, both ecologically and commercially. The carbon nanotube-loaded $CoFe_2O_4$ ($CoFe_2O_4$@CNTs) composite was made using a solvothermal technique utilizing wasted LIBs as the starting material and carbon nanotubes as support, and it was used as an efficient peroxymonosulfate (PMS, $HSO_5^-$) activator to degrade chloroquine phosphate (CQP). Scanning electron microscopy (SEM), transmission electron microscopy (TEM), an energy dispersive spectrometer (EDS), X-ray diffraction (XRD), Brunauer–Emmett–Teller (BET), and X-ray photoelectron spectroscopy (XPS) were utilized to characterize the physical and chemical properties of the catalyst generated. The impacts of $CoFe_2O_4$@CNTs dosage, PMS concentration, reaction temperature, initial pH value, starting CQP concentration, and co-existing ions have undergone extensive experimental testing. In comparison to bare $CoFe_2O_4$, the $CoFe_2O_4$@CNTs demonstrated increased catalytic activity, which might be attributed to their super electron transport capacity and large surface area. In ideal conditions, the mineralization efficiency and removal efficiency of 10 mg/L CQP approached 33 and 98.7%, respectively. By employing external magnets, the $CoFe_2O_4$@CNTs catalyst may be simply recycled and reused several times. The potential reaction mechanism in the $CoFe_2O_4$@CNTs/PMS system was also investigated. In summary, this study indicates that $CoFe_2O_4$@CNTs generated from spent lithium-ion batteries have a high potential in PMS activation for CQP and other pollutant degradation.

**Keywords:** spent lithium-ion batteries; peroxymonosulfate activation; $CoFe_2O_4$@CNTs; chloroquine phosphate; recycle

## 1. Introduction

Chloroquine phosphate (CQP) has been widely used as an antimalarial drug for several decades due to its effectiveness and tolerability against malaria [1]. CQP, in addition to being a malaria treatment, is also utilized in bladder cancer therapy and has been shown to considerably lower insulin levels in type 2 diabetes [2,3]. However, CQP is persistent, toxic, carcinogenic, and teratogenic, which can cause serious threats to the aquatic environment without proper treatment. Due to CQP's lengthy half-life, it is difficult to eradicate using traditional biological treatment procedures. As a result, large-scale utilized CQP is often dumped into ambient aquatic habitats, where it may have a cumulative effect on the biological environment and pose possible harm to humans and the entire ecosystem. As a result, developing an effective and easy approach for the treatment of CQP in water is critical.

Advanced oxidation processes based on PMS (PMS-AOPs) have become attractive alternatives for the removal of organic contaminants because of their significant oxidation capability [4–6]. Heat, UV, ultrasonic, alkaline, carbon-based materials, transition metal catalysts, and other methods have been used to activate PMS to produce reactive oxygen

species (ROS), such as sulfate radicals ($SO_4^{\bullet-}$), hydroxyl radicals ($\bullet OH$), and singlet oxygen ($^1O_2$) [7–12]. Transition metal activation has been the most widely employed of these technologies due to the benefits of moderate operating conditions and low energy usage. As $Co^{2+}$ ions and composites are so effective at activating PMS, they are widely used to remediate organic pollutants that are difficult to remove [13–15]. However, the application of a homogeneous cobalt catalyst can potentially form secondary pollution and threat both ecosystems and human health by the dissolving of $Co^{2+}$ ions [16]. Moreover, recycling and reusing Co ions from an aqueous environment is extremely challenging [17]. As a result, research into stable and reusable heterogeneous cobalt catalysts has gained prominence in recent years.

Due to its crystalline structure's stability, poor solubility, high catalytic activity, and particularly super magnetic property that is advantageous for separating from the aqueous reaction system, cobalt ferrite ($CoFe_2O_4$) could be a very promising heterogeneous candidate for PMS activation [18]. Although cobalt ferrite is efficient for activating PMS to decompose organic contaminants, its practical applications are limited due to the expensive cost of cobalt starting materials. As a result, recycling and utilizing used lithium-ion batteries as a Co source to synthesize $CoFe_2O_4$ is a very promising method for lowering the cost of $CoFe_2O_4$ production. Meanwhile, minimizing the secondary environmental damage caused by wasted lithium-ion batteries will be advantageous. In recent years, there has been a lot of attention on utilizing squandered LIBs as a Co source to synthesize Co useful materials [19–22]. Pi et al. created a composite $CoFeO_2@CN$ catalyst by activating PMS with wasted Li-ion batteries for the degradation of levofloxacin hydrochloride [21]. Liang et al. recently reported the catalytic degradation of bisphenol-A via the PMS activation of cobalt ferrite made from discarded lithium-ion batteries [23]. These produced cobalt ferrite catalysts, which demonstrated high removal efficacy for organic pollutant decomposition via PMS activation.

However, the high agglomeration propensity of magnetic cobalt ferrite may impair PMS activation performance and restrict the applicability of bare cobalt ferrite for organic pollutant degradation [24]. Hence, it is a promising and effective method to deposit cobalt ferrite on a stable support to increase the dispersion of agglomerated cobalt ferrite particles and, therefore, enhance the catalytic efficiency. Carbon nanotubes (CNTs) are an ideal option for support because of their high thermal durability, large surface area, regulated pore size, and volume [25]. Furthermore, CNTs have a high electron-transfer capacity, which is advantageous for PMS activation. CNTs alone or metal oxide/CNTs hybrid catalysts have been shown in recent years to be successful at activating PMS to degrade organic molecules [26,27].

In this study, we prepared $CoFe_2O_4@CNTs$ using hydroxy-functionalized carbon nanotubes as carriers and investigated the catalytic performance of activated PMS to degrade CQP. Meanwhile, the possible influence of various operating parameters, such as catalyst and PMS doses, environment temperature, the initial pH value, and co-existing ions, on CQP degradation was assessed. Moreover, the mechanism of CQP degradation by PMS activation with $CoFe_2O_4@CNTs$ was studied.

## 2. Results and Discussion

### 2.1. Characterization

As shown in Figure 1, XRD with a 2theta range of 20 to 70° was used to characterize the $CoFe_2O_4@CNTs$ with varied CNTs weight ratios. These five peaks at 30.1, 35.4, 43.1, 56.9, and 62.5° correspond to crystal planes of spinel-type $CoFe_2O_4$ with the standard JCPDS card number 22-1086, respectively. Meanwhile, modest CNTS peaks around 26.4° were identified in the XRD patterns, which might be attributed to interactions between $CoFe_2O_4$ and CNTs. The XRD peak shapes did not change considerably as the number of CNTs increased, showing that the $CoFe_2O_4$ nanoparticles were effectively disseminated inside the CNTs support material.

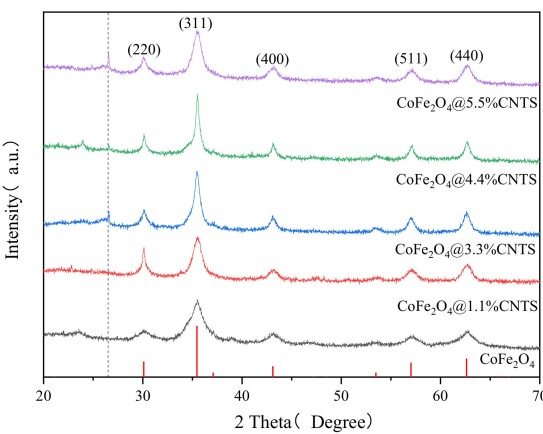

**Figure 1.** XRD patterns of the different CoFe$_2$O$_4$@CNTs composites.

As can be seen from Figure 2a, it was obvious that the CoFe$_2$O$_4$ was composed of some agglomeration of nanoparticles with irregular sizes. The surfaces of the CoFe$_2$O$_4$@CNTs composite material revealed a combination of tubes and irregular aggregates, as shown in Figure 2b. The small-sized CoFe$_2$O$_4$ particles were well dispersed on the surface of the CNTs, as revealed by TEM micrographs of the CoFe$_2$O$_4$@CNTs composite shown in Figure 2c,d, indicating that CoFe$_2$O$_4$ particles were successfully loaded onto the surface of the CNTs, which can further enhance the catalytic effect of the CoFe$_2$O$_4$@CNTs/PMS system.

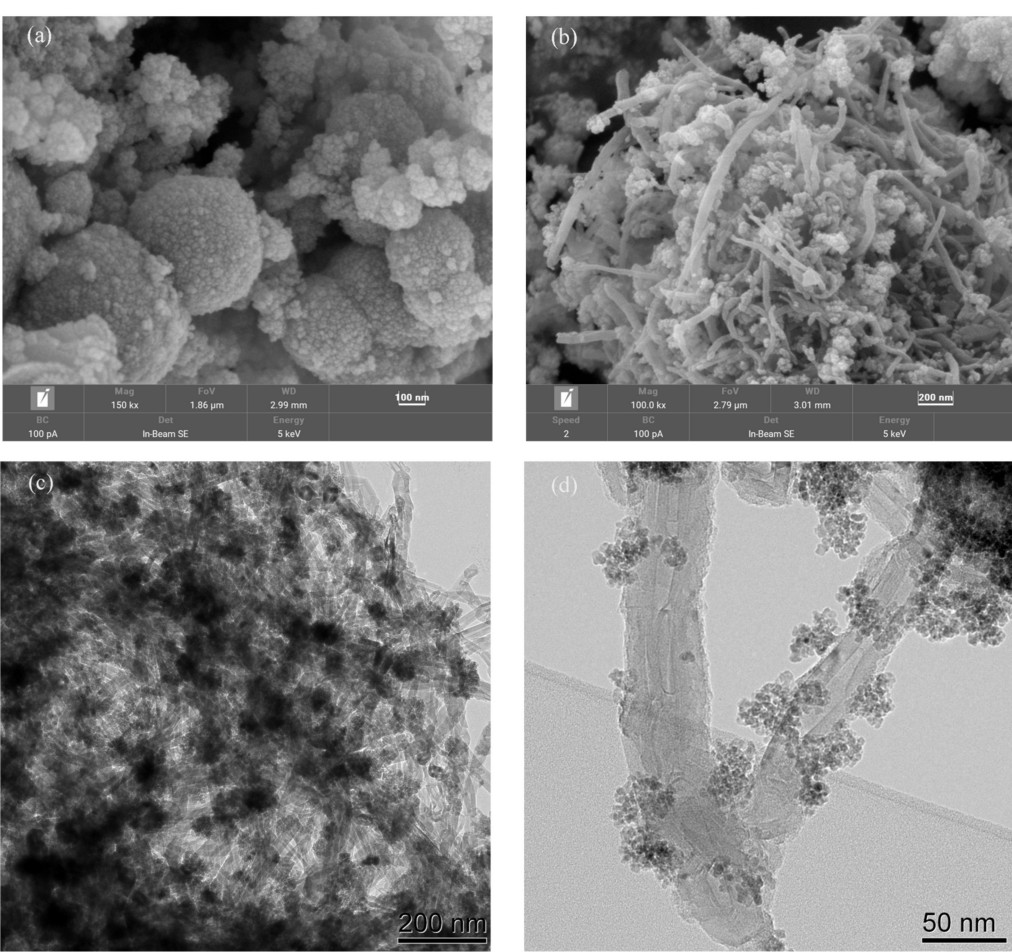

**Figure 2.** SEM images of (**a**) CoFe$_2$O$_4$, (**b**) CoFe$_2$O$_4$@CNTs, and (**c,d**) TEM images of CoFe$_2$O$_4$@CNTs.

Figure 3 depicts the EDS analysis and element mapping pictures of the CoFe$_2$O$_4$@CNTs. As EDS is a semi-quantitative analysis, it is normal that the Co and Fe atom ratio measured by EDS was 0.29, which is different from the theoretical value of 0.5. The element mapping revealed that the four key elements, Co, Fe, C, and O, were uniformly distributed on the CoFe$_2$O$_4$@CNTs, clearly increasing interaction with the PMS and so boosting catalytic effectiveness.

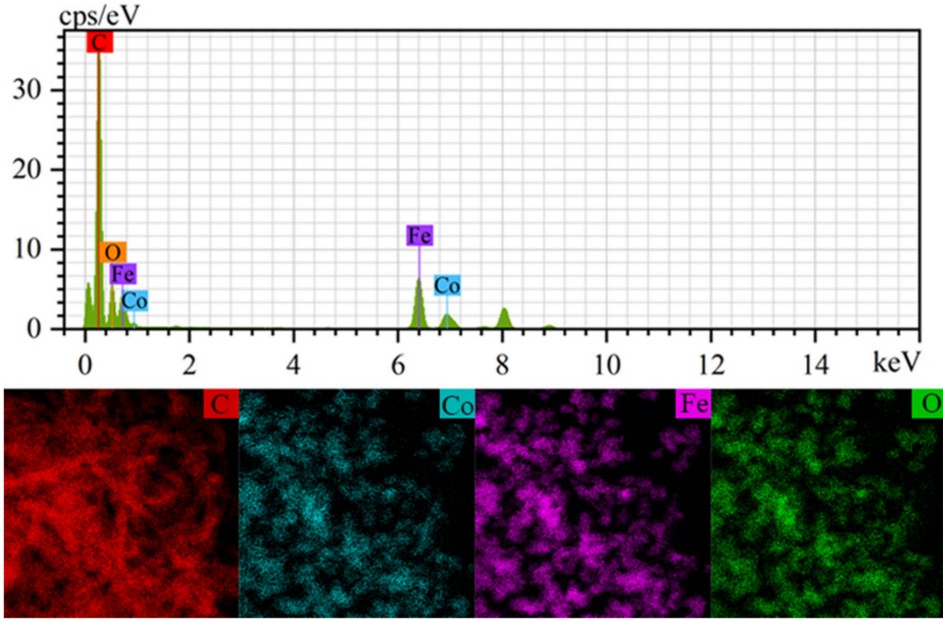

**Figure 3.** The elemental energy spectrum of CoFe$_2$O$_4$@CNTs.

The elemental composition and chemical state of CoFe$_2$O$_4$@CNTs composites were investigated using XPS analysis. As presented in Figure 4a, four elements (C, O, Fe, and Co) were observed in the full-survey XPS spectrum of the CoFe$_2$O$_4$@CNTs. As shown in Figure 4b, the three peaks with binding energies of 779.82, 781.20, and 783.25 eV can be distinguished from Co 2p3/2, which belong to Co(II) in the octahedral sites, Co(II) in the tetrahedral sites, and Co(III) in the octahedral sites, respectively. These findings are consistent with the characteristics of spinel structures [23,28]. Oct.Co(II) at 779.82 eV, Tet.Co(II) at 781.20 eV, and Oct.Co(III) at 783.25 eV accounted for 42.12, 40.61, and 17.27%, respectively. The four peaks of Fe 2p3/2 and Fe 2p1/2 and the satellite peaks between them are shown in Figure 4c. Further, the Fe$^{2+}$ peak with a binding energy of 710.3 eV and Fe$^{3+}$ peak with a binding energy of 712.49 eV were fitted by the Fe 2p3/2 spectrum [23,28,29]. The satellite peak with a binding energy of 717.51 eV between Fe 2p3/2 and Fe 2p1/2 proves the existence of Fe$^{2+}$ as well [30,31]. Fe$^{2+}$ and Fe$^{3+}$ accounted for 57.01 and 42.99%, respectively. As shown in Figure 4d, the binding energies of the three peaks fitted by the XPS spectrum of O1s are 529.5, 530.32, and 531.20 eV, respectively. According to previous literature analysis, they belong to lattice oxygen (O$_{latt}$), surface hydroxyl oxygen (O$_{OH}$), and adsorbed oxygen (O$_{ads}$), and their proportions are 39.34, 29.43, and 31.23%, respectively [23,32].

According to Figure 5, typical IV isotherms are easily visible in the pressure range of 0.6–1.0 P/P0, where H3 hysteresis loops are evident, showing the presence of mesoporous structures in the CoFe$_2$O$_4$@CNTs composites. The CoFe$_2$O$_4$@CNTs composite has an average pore size of 11.37 nm and a BET surface area of 62.5 m$^2$/g. The saturation magnetization of the CoFe$_2$O$_4$@CNTs composite is 38.30 emu/g, as determined by the magnetic hysteresis loop of CoFe$_2$O$_4$@CNTs (shown in Figure 6). Due to their strong magnetic properties, CoFe$_2$O$_4$@CNTs composites can be rapidly separated from the degradation reaction system and easily recycled and reused by applying an appropriate magnetic field, which minimizes the Co ion leaching and occurrence of secondary pollution.

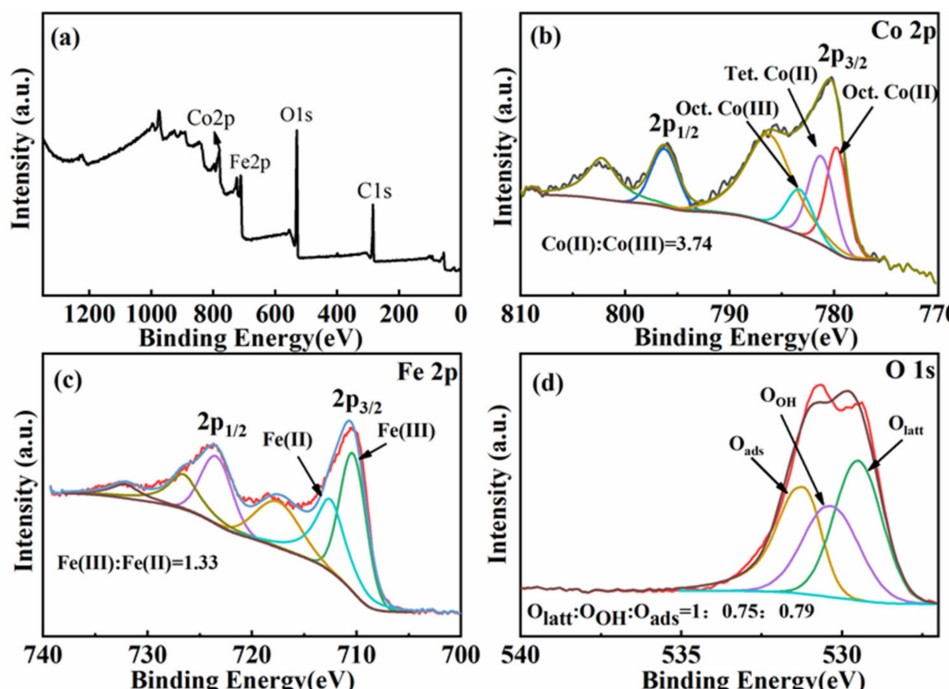

**Figure 4.** Fresh CoFe$_2$O$_4$@CNTs XPS spectra: (**a**) full-range scan of the samples, (**b**) Co 2p core level, (**c**) Fe 2p core level, and (**d**) O 1s core level.

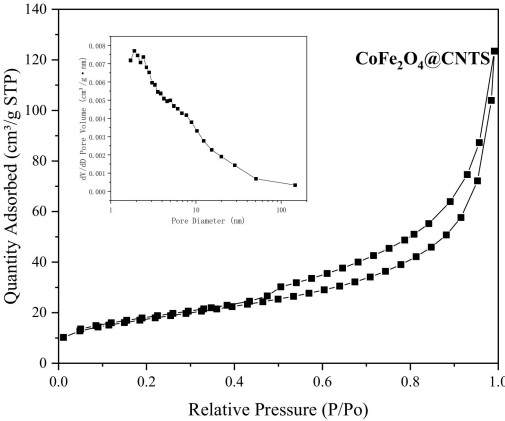

**Figure 5.** Nitrogen adsorption–desorption isotherm and pore size distribution of CoFe$_2$O$_4$@CNTs (insert).

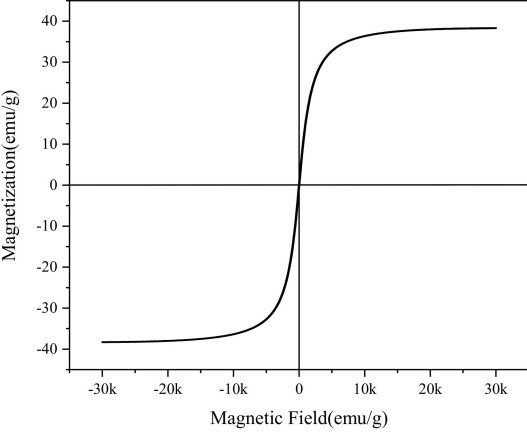

**Figure 6.** CoFe$_2$O$_4$@CNTs magnetic hysteresis loop obtained at 300 K.

### 2.2. CQP Degradation Experiments

#### 2.2.1. Effects of m(CNTs)/m(CFO) Ratios

To determine the optimal CoFe$_2$O$_4$@CNTs-based heterogeneous reaction systems and test their stability in different environments, a series of experiments were carried out to activate PMS using CoFe$_2$O$_4$@CNTs to degrade CQP. As revealed in Figure 7, the maximum removal and adsorption efficiencies of the CoFe$_2$O$_4$@CNTs and PMS for CQP were very low at 2.1 and 3.3%, respectively. When PMS was introduced to the reaction system, the removal percentage of CQP in the CoFe$_2$O$_4$@1.1%CNTs/PMS and CoFe$_2$O$_4$/PMS systems increased dramatically, reaching 93.2 and 89.6%, respectively, but CQP was only removed by 2.2% in the CNTs/PMS system. The results show that the CoFe$_2$O$_4$ content was extremely efficient for PMS activation to destroy CQP, whereas the effect of CNTs on PMS activation was relatively restricted, indicating that CoFe$_2$O$_4$ nanoparticles are the predominant active species in CoFe$_2$O$_4$@CNTs for PMS activation. As shown in Figure 7, the presence of CNTs had a beneficial influence on the catalytic activity of the CoFe$_2$O$_4$@CNTs. However, the excessive content of CNTs will reduce the performance of the catalyst. For example, the CoFe$_2$O$_4$@4.4%CNTs catalyst degraded 98.7% of CQP, whereas the CoFe$_2$O$_4$@6.6%CNTs catalyst degraded 97.4% of CQP. Figure S1 displays the degradation of CQP $K_{obs}$ by CoFe$_2$O$_4$@CNTs at various loading ratios. The $K_{obs}$ had the maximum value for the CoFe2O4@4.4%CNTs/PMS system, indicating the quickest degradation rate. As a result, CoFe$_2$O$_4$@4.4%CNTs were chosen for the following tests.

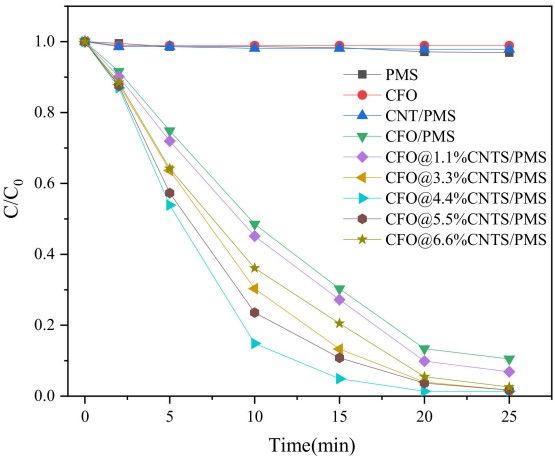

**Figure 7.** Degradation of CQP in different systems. (Experimental conditions: [[CQP]$_0$ = 10 mg/L, catalyst = 35 mg/L, PMS = 2.0 mM, initial pH = 6.5 ± 0.3, T = 25 °C.)

#### 2.2.2. Effect of Catalyst Dosage

The catalyst dosage substantially impacted the degrading efficiency of CQP because increasing the catalyst dosage increased the interaction between the CoFe$_2$O$_4$@CNTs composite and PMS and increased the catalytic active sites, resulting in more reactive species (i.e., SO$_4^{\bullet-}$, $\bullet$OH, and $^1$O$_2$). Figure 8a depicts the effect of catalyst dose on CQP degradation. The degradation rate of CQP did not improve when the catalyst dosage was raised to 35 mg/L, showing that the active site on the surface of the 35 mg/L CoFe$_2$O$_4$@CNTs was saturated. Compared with the other two catalysts for activating PMS to degrade CQP, the amount of catalyst used in this study is significantly reduced while maintaining the same catalytic efficiency (Table S1). As shown in Figure S2a, k$_{obs}$ achieved its highest value (0.215 min$^{-1}$) when the catalyst dose was 35 mg/L. As a result, the catalyst dose in future tests will be 35 mg/L.

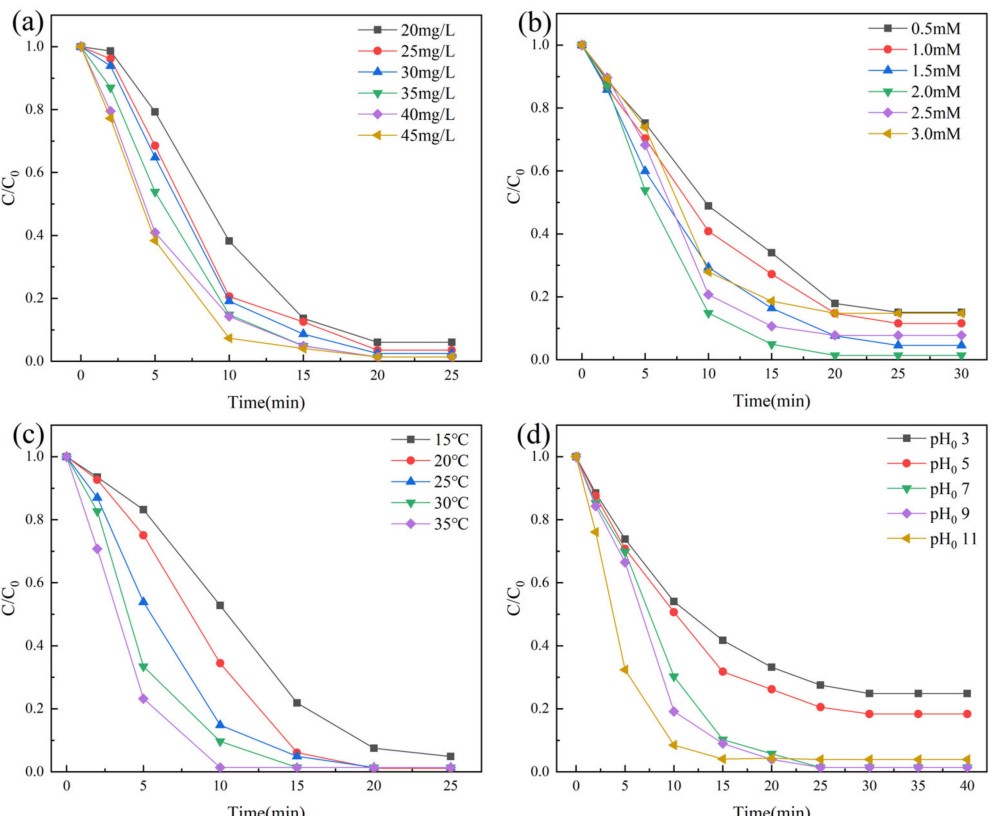

**Figure 8.** The effect of (**a**) catalyst dosage, (**b**) PMS concentration, (**c**) reaction temperature, and (**d**) initial pH values on the removal efficiency of CQP. (Experimental conditions: $[[CQP]_0 = 10$ mg/L, catalyst = 35 mg/L, PMS = 2.0 mM, initial pH = 6.5 ± 0.3, T = 25 °C.)

### 2.2.3. Effect of PMS Concentration

The effect of various PMS concentrations (0.5–3 mM) on CQP degradation was studied. As observed in Figure 8b, increasing the concentration of PMS (0.5 to 2 mM) enhanced the breakdown efficiency of CQP from 85 to 98.7%. Furthermore, as shown in Figure S2b, Kobs achieves its maximal value with a PMS of 2 mM. Nevertheless, when the PMS concentration was increased from 2 to 3 mM, the degradation of CQP was reduced. For example, when the PMS concentration was 3 mM, 85.3% of CQP was destroyed. The reason for this phenomenon is that extra $HSO_5^-$ may scavenge the created $SO_4^{\bullet-}$ or $\bullet OH$ radicals to form weaker $PMS(SO_5^{\bullet-})$ or hydroperoxyl ($HO_2\bullet$) radicals or $SO_4^{\bullet-}$ self-reactions to yield $S_2O_8^{2-}$ before CQP degradation [33]. As a result, 2 mM PMS was chosen as the best dose in further trials.

### 2.2.4. Effect of Ambient Temperature

The $K_{obs}$ of CQP degradation increased at increasing reaction temperatures (from 15 to 35 °C), as seen in Figure S2c. Nevertheless, as seen in Figure 8c, the response rate rose, but the final degradation rate did not improve considerably (95.2, 98.9, 98.7, 98.7, and 98.7%). Room temperature (25 °C) was used as the subsequent reaction condition to best imitate the degradation environment.

### 2.2.5. Effect of Initial pH

Figure 8d depicts the effect of initial pH values on CQP degradation. CQP degradation efficiency was 75.2, 81.7, 98.7, 98.6, and 97.2% at starting pH values of 3.0, 5.0, 7.0, 9.0, and 11.0, respectively. The results show that the $CoFe_2O_4@CNTs$ composite had strong catalytic activity to activate PMS in a neutral or alkaline environment, which was consistent with previous studies [33,34]. This is due to the synergistic impact of an alkaline atmosphere

combined with $CoFe_2O_4@CNTs$ on PMS activation. Large amounts of $H^+$ may consume the produced radicals and decrease the oxidation capacity of the reaction system in an excessively acidic environment. When the initial pH is in the range of 6.5–9, the efficiency of CQP degradation by the catalyst can exceed 98.6%, and the degradation rate of the catalyst is also above $0.17 min^{-1}$ in this range, as shown in Figure S2d. There is no need to change the starting pH of the solution in future trials, both economically and in terms of CQP removal impact.

### 2.2.6. Effect of CQP Initial Concentration

According to Figure 9, low starting concentrations were more favorable for CQP degradation, and between 2.5 and 5.0 mg/L, CQP could be eliminated within 15 min. Only an 86.4% degradation rate was reported after 20 min of response when the starting CQP concentration was 30 mg/L. Nevertheless, by extending the reaction time to 40 min, the pollutant of 30 mg/L may be reduced by 97.4%. The degradation efficiencies at 15 and 20 mg/L were 98.5 and 97.9%, respectively. As shown in Figure S3, CQP-degradation $k_{obs}$ dropped abruptly from 0.314 to 0.099 $min^{-1}$ as the initial CQP concentration rose from 2.5 to 30 mg/L. This is because there are a finite number of PMS and catalysts, resulting in a finite number of active species. In addition, due to the increase in CQP molecules, they compete with each other for the active species (i.e., $SO_4^{\bullet-}$, $\bullet OH$, and $^1O_2$) to lower the removal efficiency.

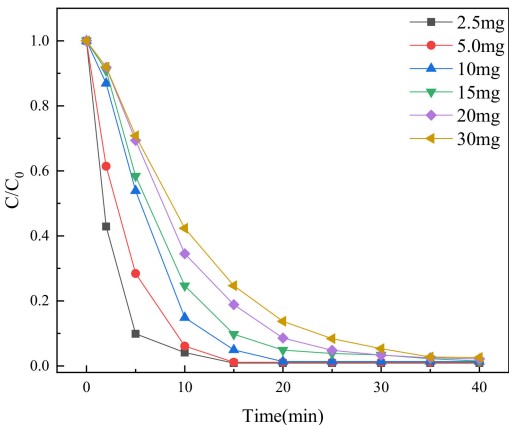

**Figure 9.** CQP degradation at various initial concentrations. (Experimental conditions: $[[CQP]_0$ = 10 mg/L, catalyst = 35 mg/L, PMS = 2.0 mM, initial pH = 6.5 ± 0.3, T = 25 °C.)

### 2.2.7. Effect of Co-Existing Ions

The anions ($Cl^-$, $HCO_3^-$, and $NO_3^-$) and humic acid (HA), which are often found in the natural aquatic environment, were chosen to evaluate their impact on the degrading efficiency of CQP. As seen in Figure 10a, HA has a modest detrimental effect on the catalytic system, with the addition of 25 mg/L HA decreasing the percentage of CQP elimination by 10.4%. As shown in Figure 10b, the coexisting $NO_3^-$ anion has essentially little negative influence on CQP degrading efficiency. The presence of $Cl^-$ anions resulted in a modest drop in CQP degradation efficiency, while 10 mM $Cl^-$ resulted in a 5.7% decrease in degradation efficiency (Figure 10c). It should also be noted that $HCO_3^-$ alone may activate PMS to destroy 18% of CQP, and the addition of $HCO_3^-$ caused a promotion impact on CQP degradation (Figure 10d) as the addition of $HCO_3^-$ causes the pH of the reaction solution to rise, therefore speeding up the process [35].

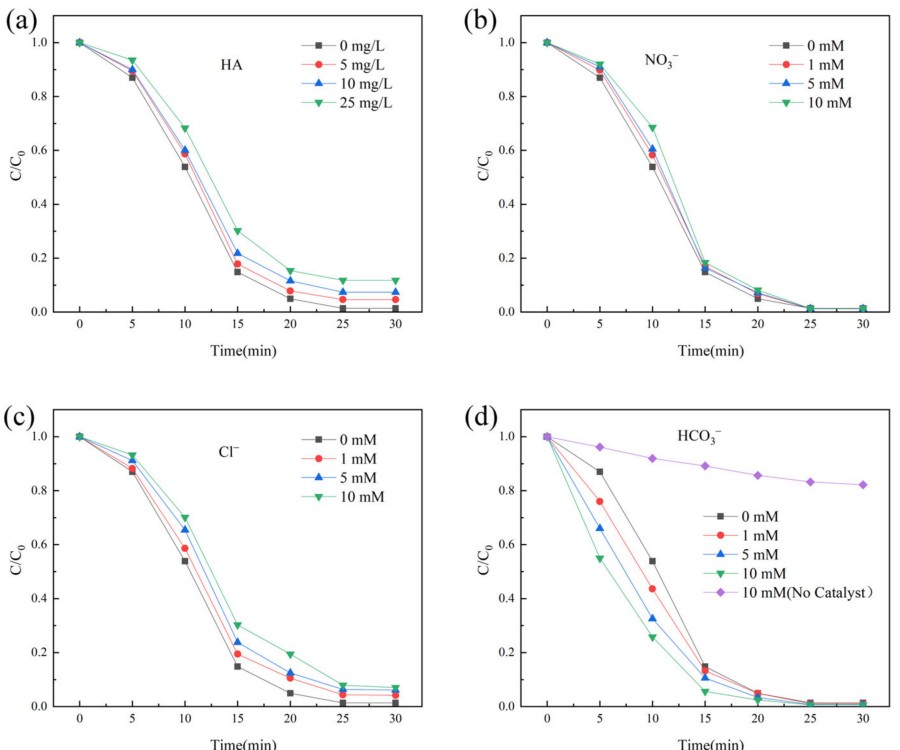

**Figure 10.** The effects of (**a**) HA, (**b**) $NO_3^-$, (**c**) $Cl^-$, and (**d**) $HCO_3^-$ on CQP degradation in the $CoFe_2O_4$@CNTs/PMS system. (Experimental conditions: $[[CQP]_0 = 10$ mg/L, catalyst = 35 mg/L, PMS = 2.0 mM, initial pH = 6.5 $\pm$ 0.3, T = 25 °C.).

## 2.3. CQP Degradation Experiments

### 2.3.1. Evaluation of Residual PMS

According to Figure 11, the highest effective PMS decomposition was achieved with the $CoFe_2O_4$@4.4%CNTs catalyst, which destroyed 97.8% of the PMS in 70 min. The PMS decomposition rates for $CoFe_2O_4$, $CoFe_2O_4$@1.1%CNTs, $CoFe_2O_4$@3.3%CNTs, $CoFe_2O_4$@5.5%CNTs, and $CoFe_2O_4$@6.6%CNTs catalysts were 88.4, 96.8, 96.6, 93, and 92.1%, respectively. These findings agreed with the CQP degradation trials.

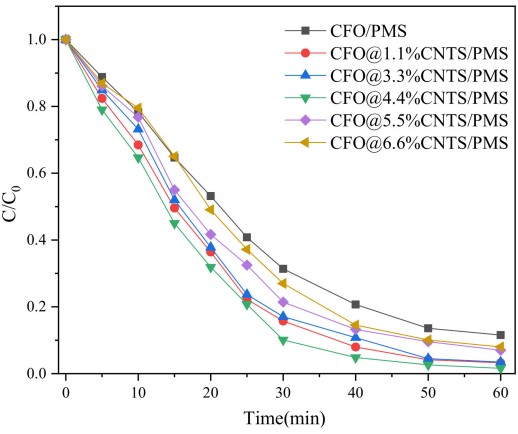

**Figure 11.** Catalytic breakdown of PMS using varied weight ratios of $CoFe_2O_4$@CNTs.

### 2.3.2. Evaluation of TOC

Total organic carbon (TOC) was evaluated to further study the CQP degrading efficiency of the CoFe$_2$O$_4$@CNTs/PMS system. As shown in Figure 12, the TOC removal effectiveness steadily rose over the first 60 min and then remained steady at 33%, indicating that part of the CQP was mineralized to inorganic compounds.

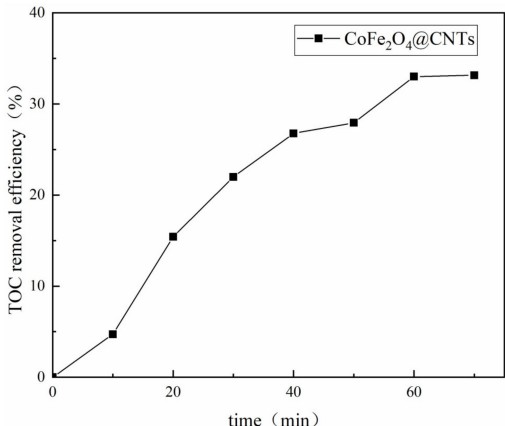

**Figure 12.** The effectiveness of TOC removal in the CoFe$_2$O$_4$@CNTs/PMS/CQP system.

### 2.3.3. Evaluation of Degradation of Different Pollutants

Degradation studies with various contaminants were carried out to further investigate the catalytic efficacy of the CoFe$_2$O$_4$@CNTs/PMS system. As shown in Figure 13, the removal efficiencies attained for rhodamine, methylene blue, methyl orange, tetracycline, doxycycline hyclate, and sulfanilamide were 100, 99.8, 99.5, 96.5, 87.4, and 49%, respectively. This indicates that the catalyst has a good degradation ability for different pollutants. In particular, rhodamine and methylene blue can be completely degraded within 10 min.

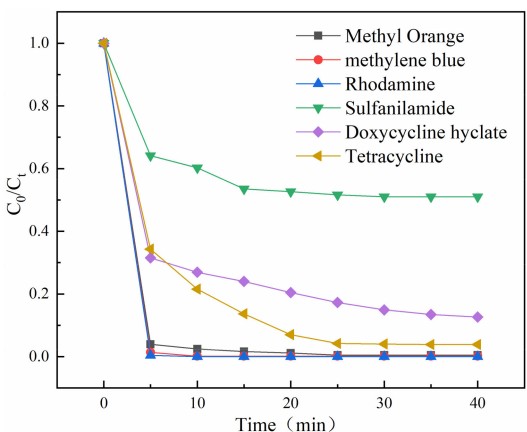

**Figure 13.** The degradation efficiency of CoFe$_2$O$_4$@CNTs/PMS system for different pollutants. (Experimental conditions: initial concentration of methyl orange, methylene blue, rhodamine, sulfanilamide, doxycycline hyclate, tetracycline = 20, 20, 20, 10, 10, 10 mg/L, PMS = 2 mM, catalyst = 35 mg/L, T = 25 °C).

### 2.3.4. Reusability and Stability of CoFe$_2$O$_4$@CNTs/PMS

Five successive reaction cycles were used to assess the recyclability and stability of the CoFe$_2$O$_4$@CNTs catalyst. To reduce catalyst mass loss, the used catalyst was cleaned three times with distilled water, three times with ethanol, and dried at 60 °C before being applied at the same dosage for the following cycle. As shown in Figure 14, CQP degradation efficiency was maintained at over 80% after five cycles. Despite the lower k$_{obs}$, the fourth cycle may degrade 98% of CQP, which is about the same as the first.

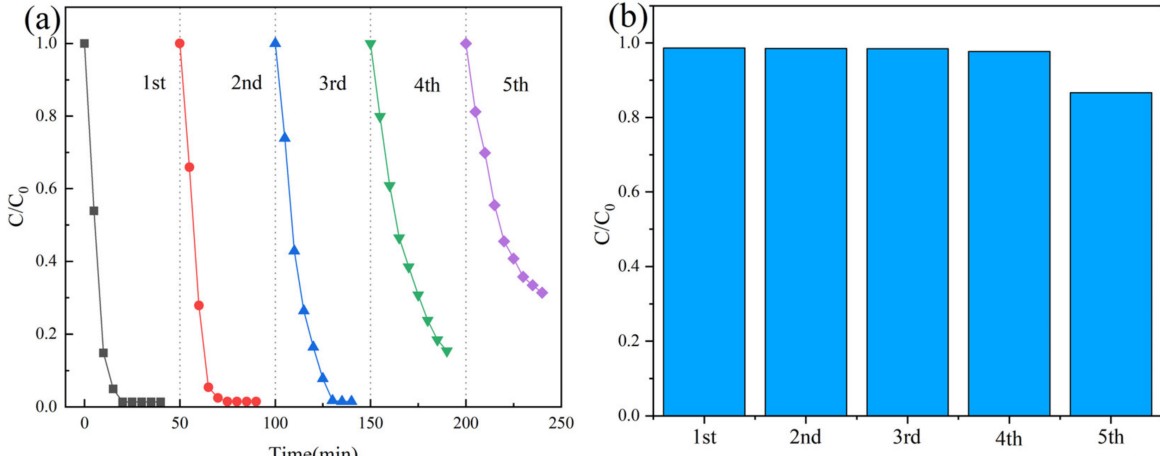

**Figure 14.** Recyclability for the degradation of CQP (**a**,**b**). (Experimental conditions: [[CQP]$_0$ = 10 mg/L, catalyst = 35 mg/L, PMS = 2.0 mM, initial pH = 6.5 ± 0.3, T = 25 °C).

In order to investigate the stability of the CoFe$_2$O$_4$@CNTs composite, the leaching amounts of cobalt and iron during the activation of PMS were tested using ICP-OES. As shown in Figure S4, compared with pure CoFe$_2$O$_4$ synthesized from spent LIBs (193.38 and 35.73 μg/L), the leaching amounts of cobalt and iron of CoFe$_2$O$_4$@CNTs significantly decreased to 78.01 and 17.11 μg/L, respectively. This is due to the high surface area and strong adsorption capacity of CNTs, which can effectively limit the dissolution and diffusion of cobalt ions. In addition, due to the good electrical conductivity of CNTs, the electrocatalytic activity of the composite can be improved, thus promoting the degradation of CQP and contributing to the stability of CoFe$_2$O$_4$.

## 2.4. CQP Degradation Experiments

### 2.4.1. Species Identification in the CoFe$_2$O$_4$@CNTs/PMS System

The free radical quenching test was used to identify the active species in the CoFe$_2$O$_4$@CNTs/PMS system. According to prior research, the most active species created by the PMS activation process were SO$_4^{\bullet-}$ and $\bullet$OH [21,36]. In this study, methanol (Meth) was selected to scavenge both SO$_4^{\bullet-}$ and $\bullet$OH because of the reported large quenching rate constant of $3.2 \times 10^6$ and $9.7 \times 10^8$ M$^{-1}$ s$^{-1}$, respectively [37]. Tert-butyl alcohol (TBA) was used in this work to quench $\bullet$OH as its reaction rate constant with $\bullet$OH (($3.8$–$7.6) \times 10^8$ M$^{-1}$ s$^{-1}$) is substantially quicker than with SO$_4^{\bullet-}$ (($4$–$9.1) \times 10^5$ M$^{-1}$ s$^{-1}$) [38]. Meanwhile, p-benzoquinone (BQ) was further chosen to quench O$_2^{\bullet-}$, with a rate constant of $(0.9$–$1.0) \times 10^9$ M$^{-1}$s$^{-1}$ [39]. As shown in Figure 15a, when MeOH, TBA, and BQ were added to the CQP solution, the CQP removal efficiency declined from 98.7 to 15.6%, 97.1 and 28.0%, respectively, demonstrating that SO$_4^{\bullet-}$ and O$_2^{\bullet-}$ had a substantial influence, while $\bullet$OH had little effect on the CQP solution deterioration.

In addition, L-histidine was used in this work to verify the non-radical process because it is a well-known typical quencher for $^1$O$_2$ [40]. As revealed in this work, the inhibition ratio of CQP accounted for 63% when 1 mM L-histidine was added to the reaction solution. Based on the results mentioned above, both the radical (SO$_4^{\bullet-}$, $\bullet$OH, and O$_2^{\bullet-}$) and non-radical ($^1$O$_2$) pathways were involved in the CQP degradation process with the CoFe$_2$O$_4$@CNTs/PMS system.

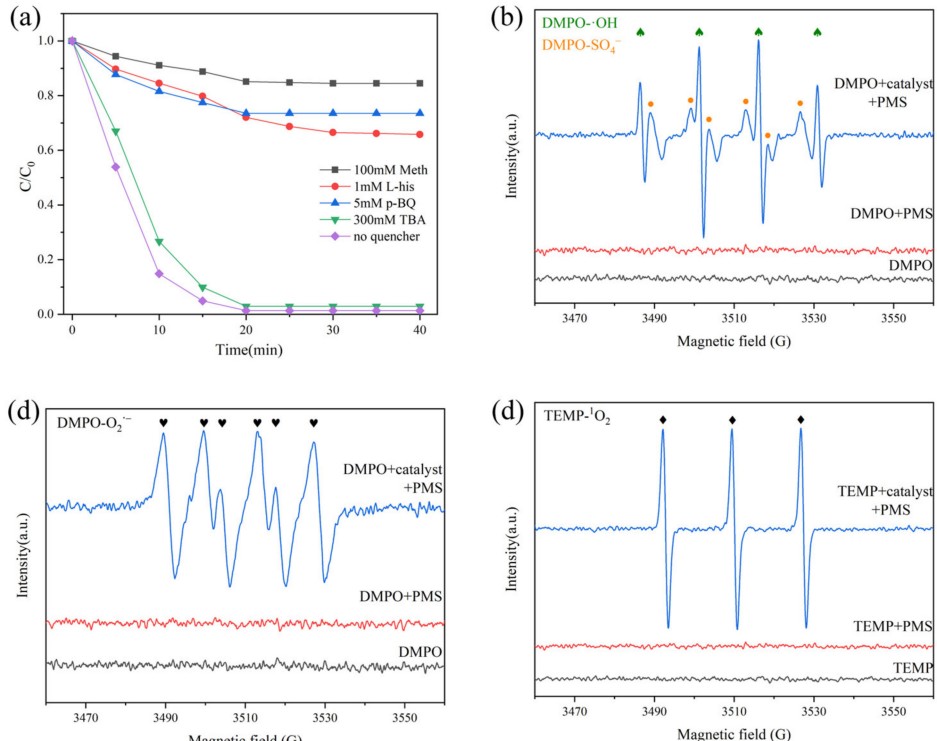

**Figure 15.** (**a**) Effect of radical scavengers on the degradation of CQP. EPR spectra in CoFe$_2$O$_4$@CNTs/PMS and PMS systems with DMPO (**b**,**c**) and TEMP (**d**) as the trapping agent. (Experimental conditions: [[CQP]$_0$ = 10 mg/L, catalyst = 35 mg/L, PMS = 2.0 mM, initial pH = 6.5 ± 0.3, T = 25 °C.)

To further determine the reactive species produced by the oxidation of CQP by PMS activation, radical trapping tests were carried out using EPR technology to further identify the reactive species generated by the oxidation reaction of CQP by PMS activation. The spin-trapping agent for SO$_4^{\bullet-}$, •OH, and O$_2^{\bullet-}$ was 5,5-dimethyl-1-pyrroline N-oxide (DMPO), and the trapping agent for $^1$O$_2$ was 2,2,6,6-tetramethylpiperidine (TEMP). As shown in Figure 15b,c, there were no obvious characteristic peaks in the sole DMPO and DMPO/PMS systems. The typical quartet peaks of the DMPO-•OH adduct, as well as the sextet peaks of the DMPO-SO$_4^{\bullet-}$ and DMPO-O$_2^{\bullet-}$ adducts, were observed, indicating the presence of SO$_4^{\bullet-}$, •OH, and O$_2^{\bullet-}$ during the catalytic degradation process over the CoFe$_2$O$_4$@CNTs/PMS system. As expected, strong TEMP-$^1$O$_2$ signals were observed in the CoFe$_2$O$_4$@CNTs/PMS system, as shown in Figure 15d, whereas no significant characteristic peaks were observed in the individual TEMP or TEMP/PMS systems. The signals were due to the adduct of TEMP, with $^1$O$_2$ generated from the CoFe$_2$O$_4$@CNTs/PMS system; it has also been reported by other groups that the carbon-containing materials can activate PMS to generate $^1$O$_2$ by non-radical mechanisms [41].

### 2.4.2. XPS Characterizations of CoFe$_2$O$_4$@CNTs during the Reaction Cycles

To better explore the activation mechanism of PMS by CoFe$_2$O$_4$@CNTs, the XPS characterizations of fresh, after one reaction, and after five reactions of CoFe$_2$O$_4$@CNTs were carried out. As shown in Figure 16a, after one reaction, the relative content of Co(III) increased from 17.27 to 25.12%, which proves that Co(II) provides an electron for the activation of PMS and produces Co(III), and the relative content of Co(II) was reduced from 42.12 to 38.26% (Oct.Co(II)) and 40.61 to 36.62% (Tet. Co(II)), respectively. After five cycles of reaction, the proportion of Co(II) in the octahedral and tetrahedral positions decreased to 27.11 and 15.74%, and the proportion of Co(III) increased to 57.16%, indicating that Co(II) might donate the electrons to activate PMS, leading to the increase in Co(III).

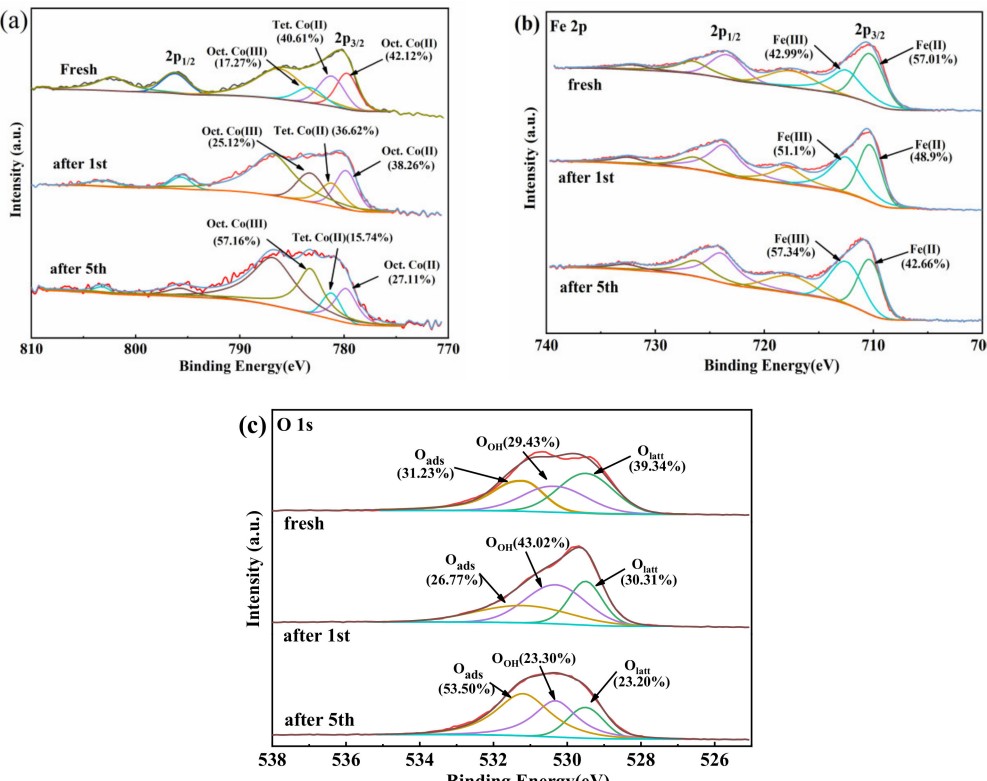

**Figure 16.** (**a**) Co 2p, (**b**) Fe 2p, and (**c**) O 1s XPS spectra of $CoFe_2O_4$@CNTs before and after reaction.

In addition, as shown in Figure 16b, the relative content of Fe(II) also decreased after the reaction. After one reaction, the relative content of Fe(II) decreased from 57.01 to 48.9%, and after five cycles of reaction, the relative content of Fe(II) decreased to 42.66%. This is because the reaction of activating PMS also involves Fe(II) losing electrons and being oxidized to Fe(III). This is consistent with the previous report that the higher the relative content of $Co^{2+}$ and $Fe^{2+}$ in $CoFe_2O_4$, the stronger the activation ability of PMS [42].

Figure 16c shows the XPS spectra for O1s before and after the reaction. Before the reaction, the proportions of lattice oxygen ($O_{latt}$), surface hydroxyl oxygen ($O_{OH}$), and adsorbed oxygen ($O_{ads}$) were 39.34, 29.43, and 31.23%, respectively. After one cyclic reaction, $O_{OH}$ increased to 43.02% and decreased to 23.3% after five cyclic reactions. The $O_{OH}$ shows a trend of increasing first and then decreasing, probably because the activity of the catalyst in the first reaction is the best, so the activation of PMS is also the best, with more hydroxyl compounds formed on the surface of the catalyst. With the increase in the cycling numbers, the catalytic activity of the catalyst becomes worse, the generated hydroxyl groups become less, and the surface hydroxyl compounds also decrease.

### 2.4.3. Electrochemical Measurement Analysis

To further explore the effect of electron transfer and the electron migration rate of $CoFe_2O_4$@CNTs on the catalytic activation of the PMS system, an electrochemical study was carried out. According to previous reports, $CoFe_2O_4$@CNTs possess good electrochemical performance and excellent potential to be electrode material [43]. Meanwhile, the electron-transfer ability also significantly impacts the catalyst-activated PMS system [33]. As shown in Figure 17a, the EIS Nyquist plot of $CoFe_2O_4$@CNTs was smaller and revealed its lower charge-transfer resistance and stronger electron-transfer ability.

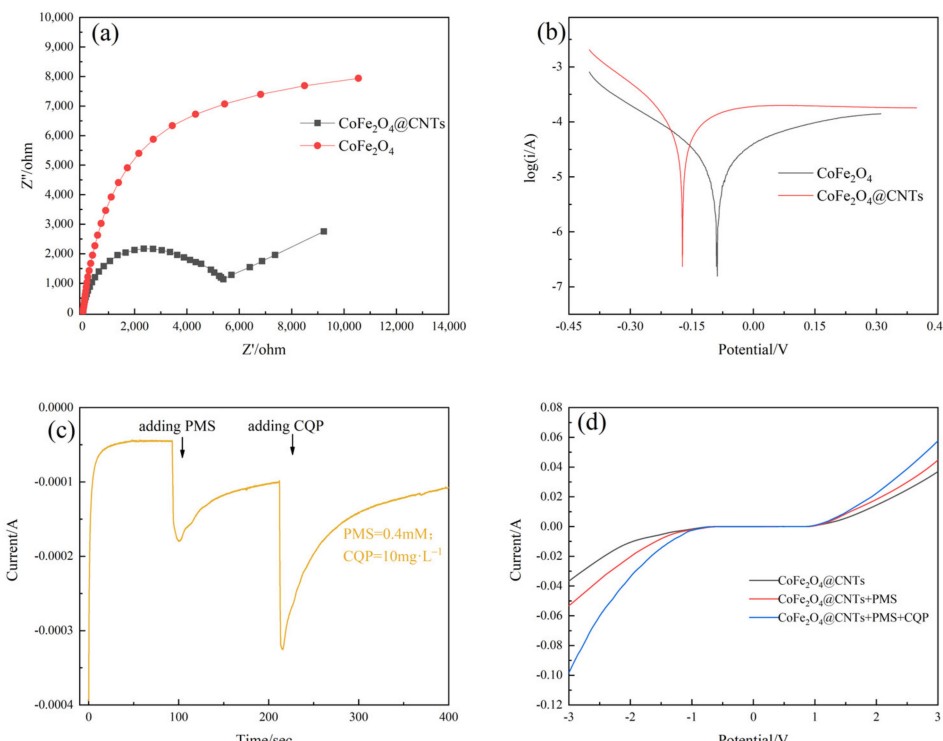

**Figure 17.** (**a**) EIS Nyquist plots of CoFe$_2$O$_4$ and CoFe$_2$O$_4$@CNTs; (**b**) Tafel polarization curves of CoFe$_2$O$_4$ and CoFe$_2$O$_4$@CNTs composite; (**c**) the current curve measurements upon the addition of PMS and CQP with CoFe$_2$O$_4$@CNTs; (**d**) the linear sweep voltammetry (LSV) of CoFe$_2$O$_4$@CNTs, CoFe$_2$O$_4$@CNTs/PMS and CoFe$_2$O4@CNTs/PMS/CQP systems.

In addition, the corrosion potential of the catalyst was measured using the Tafel polarization curve, and the corrosion current value was further calculated using the electrochemical workstation. As displayed in Figure 17b, the CoFe$_2$O$_4$@CNTs catalyst ($9.637 \times 10^{-5}$ A) had a higher corrosion current compared with the pure CoFe$_2$O$_4$ catalyst ($1.899 \times 10^{-5}$ A), which correspondingly suggests that the CoFe$_2$O$_4$@CNTs had a higher electron-transfer rate. As shown in Figure 17c, the current I−t curve was recorded to monitor the charge transfer. When PMS was injected into the system at 100 s, a negative current immediately appeared. After CQP was injected at 200 s, a more noticeable change in current was generated. This indicates that CoFe$_2$O$_4$@CNTs can promote PMS decomposition and CQP degradation through electron transfer [44].

As shown in Figure 17d, the linear sweep voltammetry (LSV) curve of CoFe$_2$O$_4$@CNTs/ PMS is higher than that of the bare CoFe$_2$O$_4$@CNTs system, which may be due to the fact that PMS molecules bind to the catalyst as a metastable reaction complex. In addition, adding CQP to the CoFe$_2$O$_4$@CNTs/PMS system will cause CQP to be oxidized by the catalyst attached to the electrode surface, so the CoFe$_2$O$_4$@CNTs/PMS/CQP curve is highest.

### 2.4.4. Possible Reaction Mechanism

Based on the above analysis on quenching, EPR, XPS, and electrochemical, the ROS reaction and electron-transfer process were both identified as the key processes for CQP degradation by CoFe$_2$O$_4$@CNTs/PMS. As shown in Figure 18, first, for the radical process, M(II) (Co(II), and Fe(II)) on the catalyst surface rapidly combines with water molecules to form surface hydroxyls, and these M(II)-OH rapidly combine with PMS to form [≡M(II)-OH-OSO$_3$]$^+$ complexes, which then produce SO$_4$$^{\bullet-}$ by one-electron transfer. SO$_4$$^{\bullet-}$ consume H$_2$O to produce •OH. In addition, O$_2$$^{\bullet-}$ can be formed from the conversion of •OH:

$$HSO_5^- \rightarrow SO_5^{2-} + H^+ \tag{1}$$

$$SO_5^{2-} + H_2O \rightarrow SO_4^{2-} + H_2O_2 \tag{2}$$

$$H_2O_2 + \bullet OH \rightarrow HO_2\bullet- + H_2O \tag{3}$$

$$HO_2^{\bullet-} \rightarrow O_2^{\bullet-} + H^+ \tag{4}$$

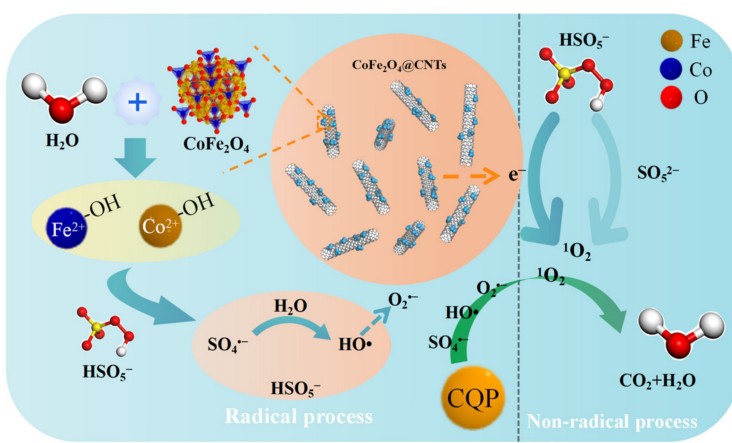

**Figure 18.** Reaction mechanism for the degradation of CQP in CoFe$_2$O$_4$@CNTs/PMS system.

For the non-radical process, the CQP contaminant would be absorbed to the surface of the CoFe$_2$O$_4$@CNTs, and the electron would transfer from the CQP contaminant to the PMS to promote the generation of $^1O_2$. In addition, the self-decomposition of PMS can also create $^1O_2$. Finally, the strong conductivity of the material and the electron-transfer process greatly promote the reaction efficiency of the degradation system:

$$SO_5^{2-} + HSO_5^- \rightarrow {}^1O_2 + 2SO_4^{2-} + H^+ \tag{5}$$

## 3. Materials and Methods

### 3.1. Chemicals and Material

The used Li-ion batteries from Bluetooth speakers were recycled (SAMSUNG ICR18650-22F). Chloroquine diphosphate salt (CQP, 98%), hydroxy purified multi-walled carbon nanotubes (>95%, ID: 5–15 nm, OD: >50 nm, length: 10–20 μm, −OH: 0.71wt%), peroxymonosulfate (KHSO$_5$·0.5KHSO$_4$·0.5K$_2$SO$_4$, PMS), tert-butanol (TBA) (C$_4$H$_{10}$O, AR, ≥99.0%), L-Histidine (C$_6$H$_9$N$_3$O$_2$, 99.5%), and citric acid (C$_6$H$_8$O$_7$, AR, ≥99.5%) were purchased from Shanghai Macklin Biochemical Technology Co., LTD (Shanghai, China). Ethanol (AR), ammonia solution (NH$_3$.OH, 25%, AR), and iron(III) sulfate hydrate (Fe$_2$(SO$_4$)$_3$·XH$_2$O, AR) were purchased from Tianjin Kermel Chemical Reagent Co., LTD (Tianjin, China). Methanol (GC, ≥99.9%), and 2-2′-azinobis (3-ethylbenzthiazoline-6-sulfonic acid) (ABTS) were purchased from Aladdin Chemical Reagent Co., Ltd. 1,4-benzoquinone (98%) was purchased from Jiuding Chemical (Shanghai, China) Technology Co., LTD. Sulfuric acid (H$_2$SO$_4$, 98%) and H$_2$O$_2$ (30%) were purchased from Fuchen (Tianjin, China) Chemical Reagent Co., LTD. Sodium hydroxide (NaOH, AR) was purchased from Chaoyang Hecheng Chemical Reagent Co., LTD (Beijing, China). All compounds were utilized without being purified further.

### 3.2. Synthesis of CoFe$_2$O$_4$ and CoFe$_2$O$_4$@CNTs

The spent lithium battery (ICR18650-26F) was immersed in a 1 M NaCl solution for 24 h to eliminate the remaining electric amount and prevent the battery from short-circuiting and spontaneously combusting. The battery was then physically dismantled after adequate safety precautions were taken, and positive and negative electrode materials

were gathered for usage. To improve the Co content of the anode material and eliminate impurities, it was first heated at 500 °C for 1 h to burn the binder and organic additives. The anode material was then scraped off the aluminum foil with a scraper and heated for 1 h at 800 °C to remove carbon and unburned organic debris [45]. The $CoFe_2O_4$ preparation process using spent lithium-ion batteries was according to the modified method initially reported by Yao et al. [46]. The separated anode materials were then dissolved in a 50 g/L solid-to-liquid solution of $H_2SO_4$ and $H_2O_2$ (V/V = 1:9). The solution comprising $Co^{2+}$, $SO_4^{2-}$, and other ions was obtained by filtering the mixture and separating the filtrate. According to the results of ICP-OES (Table S2), the ratio of $Co^{2+}$ and $Fe^{3+}$ content in the filtrate was adjusted to 1:2. A modified version of solvothermal methods was used to manufacture composites of $CoFe_2O_4$@CNTs with varying CNTs-loading ratios [21,46].

In the actual synthesis process, 0.12 g CNTs was introduced to 80 mL water and ultrasonically treated for 60 min. The necessary amount of leaching solution and $Fe_2(SO_4)_3$ was then added to 80 mL of water to achieve Co and Fe concentrations of 0.1 and 0.2 M, respectively. Following that, the two solutions were combined, and citric acid was added to the solution in an amount equal to the total number of metal ions. The pH was then corrected to 8.0 using $NH_3 \cdot H_2O$, and the solution was then agitated for 10 h in a water bath mixer at 60 degrees Celsius until it turned gelatinous. Lastly, the gelatinous solution was moved to a stainless-steel autoclave lined with Teflon and kept at 200 °C for 24 h. Upon completion, the sediment in the reactor was separated and washed three times with deionized water and anhydrous ethanol.

For convenience, with various mass fractions of CNTs (0, 1.1, 3.3, 4.4, 5.5, 6.6%), the resulting products were named CFO, $CoFe_2O_4$@1.1%CNTs, $CoFe_2O_4$@3.3%CNTs, $CoFe_2O_4$@4.4%CNTs, $CoFe_2O_4$@5.5%CNTs, and $CoFe_2O_4$@6.6%CNTs.

### 3.3. Analytical Methods

TESCAN MIRA LMS was used to obtain images of scanning electron microscopy (SEM) and energy-dispersive X-ray spectroscopy (EDS). X-ray diffraction (XRD) examination was carried out utilizing an outfitted SmartLab X-ray diffractometer. The Tecnai G2 F30 S-TWIN was used for transmission electron microscopy (TEM). The pore size distribution and specific surface area (SSA) of the Brunauer–Emmett–Teller (BET) method were measured on an ASAP 2460 Version 3.01 surface area analyzer. To evaluate X-ray photoelectron spectroscopy (XPS), a Thermo Scientific ESCALAB 250Xi spectrometer was employed. QUANTUM DESIGN PPMS-9 (vibrating sample magnetometer) was used to measure the magnetic characteristics of $CoFe_2O_4$@CNTs at room temperature. The iron and cobalt ion concentrations were determined using the Agilent 7800 ICP-OES. CQP concentration was determined by UV-spectrophotometric analysis on Mapada P4. Bruker A300-10/12 spectrometer was used to detect active free radicals using electron paramagnetic resonance (EPR). Shimadu TOC-L was used to quantify total organic carbon (TOC).

All batch tests were performed in duplicate in 250 mL beakers using a Cyclotron oscillator. The pH was adjusted using 1.0 mM $H_2SO_4$ or NaOH. In a typical test, certain amounts of catalyst and PMS were dispersed into an aqueous solution containing desired amounts of CQP to motivate the reaction. Throughout the reaction, the reaction solution was withdrawn at predetermined intervals and transferred to a test tube containing 0.2 mL of methanol, which was used to quench the remaining PMS and then filtered through a PTFE (0.22 μm) membrane before organic analysis.

$CoFe_2O_4$@CNTs were collected using a magnet, completely cleaned with deionized water and ethanol, and dried in a vacuum drying oven at 60 °C for the recyclability experiments. The method described by Liu et al. was used to obtain the amount of PMS residue in the solution [47]. The pseudo-first-order kinetic pattern was used to characterize the catalytic oxidation rate of the CQP, as seen in the equation below [48]:

$$Ln(C/C_0) = -K_{obs}t \tag{6}$$

where $K_{obs}$ (min$^{-1}$) is the pseudo-first-order rate constant of CQP degradation, t (min) is the reaction time, C (mg/L) is the CQP concentration at time (t), and $C_0$ is the starting CQP concentration.

## 4. Conclusions

CoFe$_2$O$_4$@CNTs catalysts were generated in this work utilizing spent lithium-ion batteries as starting materials and CNTs as support. The CoFe$_2$O$_4$@CNTs catalyst was highly efficient in activating PMS for the removal of CQP. The greatest CQP degradation efficiency was obtained in the optimum reaction conditions of a 35 mg/L CoFe$_2$O$_4$@4.4%CNTs catalyst dosage, 2 mM PMS concentration, and starting pH of 6.5, with 98.7% CQP removal efficiency and 33% mineralization efficiency. The electron transport performance of the CoFe$_2$O$_4$@CNTs composite is better than that of bare CoFe$_2$O$_4$ particles. Free radical quenching test, EPR, and XPS analysis results confirmed that SO$_4^{\bullet-}$, O$_2^{\bullet-}$, and $^1$O$_2$ were the dominant active species in the CoFe$_2$O$_4$@CNTs/PMS system for PMS activation. A reaction mechanism was proposed to explain the formation and contribution of reactive radicals as well as the synergetic effects between Co and Fe. Moreover, the CoFe$_2$O$_4$@CNTs/PMS system showed good efficacy against additional pollutants such as rhodamine, methylene blue, methyl orange, and tetracycline. Furthermore, the produced CoFe$_2$O$_4$@CNTs were shown to be stable and reusable, with the ability to be reused for at least four cycles with extremely minimal cobalt release. In summary, CoFe$_2$O$_4$@CNTs produced from spent lithium-ion batteries can be potential heterogeneous catalysts in PMS activation for the removal of CQP and other contaminants.

**Supplementary Materials:** The following supporting information can be downloaded at: https://www.mdpi.com/article/10.3390/catal13040661/s1, Figure S1: K$_{obs}$ of CQP degradation in different systems; Figure S2: The K$_{obs}$ under reaction conditions of different (a) catalyst dosage, (b) PMS concentration, (c) reaction temperature, and (d) initial pH values on the removal efficiency of CQP.; Figure S3: K$_{obs}$ at various initial CQP concentrations; Table S1: Comparison of the reaction parameters with previously reported catalysts for PMS activation; Table S2: Concentration of key elements in leaching solution using ICP-OES.

**Author Contributions:** Conceptualization, Z.H. and B.S.; methodology, Z.H., P.Y. and B.S.; validation, Z.H. and B.S.; formal analysis, J.L. and S.X.; investigation, Z.H. and J.L.; resources, Z.H., S.G., P.Y. and B.S.; data curation, S.G., J.L. and S.X.; writing—original draft preparation, Z.H. and J.L.; writing—review and editing, Z.H., S.X. and X.T.; visualization, S.X. and S.G.; supervision, P.Y. and B.S.; project administration, S.G., P.Y. and B.S.; funding acquisition, Z.H. and B.S. All authors have read and agreed to the published version of the manuscript.

**Funding:** This research was funded by the Joint Funds of the National Natural Science Foundation of China (U20A20302), the Overseas High-level Talents Introduction Plan Foundation of Hebei Province (E2019050012), the Innovative Group Projects in Hebei Province (E2021202006), and the Fundamental Research Foundation of the Hebei University of Technology (JBKYTD2001).

**Data Availability Statement:** No data were used for the research described in the article.

**Conflicts of Interest:** The authors declare no conflict of interest.

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
