# Peer review of "Activation of Peroxymonosulfate Using Spent Li-Ion Batteries for the Efficient Degradation of Chloroquine Phosphate"

_catalysts, doi:10.3390/catal13040661_

Round 1

Reviewer 1 Report

In this manuscript, a carbon nanotubes loaded CoFe2O4 was prepared by using spent Li-ion batteries and used for persulfate activation. It is a systematic and interesting study with a lot of characterization and analysis. There are some suggestions to further improve this work.

1.     How is the ratio of CoFe2O4 on the CoFe2O4@CNTs?

2.     Line 247, what is the pH after HCO3- addition?

3.     Part 2.3.4, the title does not match the content.

4.     What is the concentration of Fe or Co in the solution? 

Author Response

We would like to first thank you for your time, and input.  we trust that we have addressed all the comments, questions, and suggestions point-by-point in the attachment. Please see the attachment.

Reviewer 2 Report

This manuscript produced a CoFe2O4@CNTs catalyst for the removal of CQP by using spent lithium-ion batteries as starting materials and CNTs as support. Comprehensive characterization was carried out for the material properties and the catalysis performance. However, to improve the current manuscript, I believe the following comments should be addressed:

Comments 1): Full names of acronyms are missing on lines 15, and 42. Typos need to be corrected on line 46. Many similar typos are seen in the manuscript and need to be corrected.

Comments 2): The experimental details need to be added. XPS, for example, what is the calibration standard of XPS binding energy? Which standard is followed? ASTM E2108-16 standard (DOI: 10.1520/E2108-16) or any other standard? Considering that the main conclusion of this work is supported by the XPS results, before drawing any conclusion from XPS, the binding energy has to be carefully calibrated. Note that the C 1s calibration method is highly arbitrary, which results in incorrect spectral interpretation, contradictory results, and generates a large spread in reported BE values for elements even present in the same chemical state. (see G. Greczynski et al., Progress in Materials Science, Volume 107, January 2020, 100591, https://doi.org/10.1016/j.pmatsci.2019.100591)

Similarly, details need to be added for all the characterization techniques.

Comments 3): Is the elemental dosage composition, 1.1%, 3.3%, etc., consistent with the characterization result, XPS, and EDX? Does the CNT account for 1.1%, 3.3%, etc. of the catalyst?

Author Response

(The authors gave the same response as above.)
